# Community knowledge, attitudes, and practices toward cutaneous leishmaniasis in central Ethiopia

Tagesech Yohannes[1,2]*, Getenet Beyene[2], Teketel Ermias Geltore[3], Daniel Geleta[4], Zeleke Mekonnen[2]

**1** Department of Medical Laboratory Science, School of Medical Laboratory Sciences, Institute of Health, Wolaita Sodo University, Sodo, Ethiopia, **2** Department of Medical Laboratory Science, Faculty of Health Sciences, Jimma University, Jimma, Ethiopia, **3** Department of Midwifery, College of Medicine and Health Science, Wachamo University, Durame Campus, Durame, Ethiopia, **4** Department of Epidemiology and Biostatistics, Faculty of Public Health, Jimma University, Jimma, Ethiopia

* tagutafe@gmail.com

## Abstract

### Background

Cutaneous leishmaniasis is the most prevalent clinical form of leishmaniasis and remains a significant public health problem, particularly in low- and middle-income countries where it disproportionately affects disadvantaged populations. The disease causes a far-reaching impact to the community where community-level awareness of its transmission, prevention, and treatment often remains low and less studied. Therefore, the present study aimed to assess community knowledge, attitudes, and practices related to CL in the Kambata Zone of Central Ethiopia.

### Methods

Ethics statement: Ethical approval was granted by the Institutional Review Board (IRB) of Jimma University with the reference number JUIH/IRB/684/23. Authorization was also obtained from the Kambata zone Health Bureau to proceed with the study. All participants provided written informed consent in accordance with the principles outlined in the Helsinki Declaration. To uphold confidentiality, participant identities were anonymized, and data was securely stored in a separate location at the Principal Investigator's office. Community-based cross-sectional study design was employed from February 2024 to August 2024 in Kambata zone to assess knowledge, attitude, and practice of community toward cutaneous leishmaniasis. Systematic sampling technique was used to select individuals. A pretested structured questionnaire was used to collect data from the household head. Bivariate and multivariate logistic regression analyses were performed to determine the relationship between the participants' knowledge, attitude, and practice toward cutaneous

**Data availability statement:** All relevant data are within the manuscript and its Supporting information files.

**Funding:** The author(s) received no specific funding for this work.

**Competing interests:** The authors have declared that no competing interests exist.

leishmaniasis and socio demographic characteristics. The association between the independent and outcome variables was presented in the form of a table showing the adjusted odds ratio (AOR) along with their 95% confidence interval (CI). The level of statistical significance was declared at P-value < 0.05.

## Results

Out of 387 participants, 72.9% were male and 66.1% resided in rural areas. Only 18.1% had good knowledge, 13.2% a positive attitude, and 11.6% good practices toward cutaneous leishmaniasis. Good knowledge of cutaneous leishmaniasis was significantly associated with age 18–40 years (AOR = 4.47; 95% CI: 2.07–9.67), male sex (AOR = 0.25; 95% CI: 0.11–0.56), formal education (AOR = 3.7; 95% CI: 2.18–8.75), family history of CL (AOR = 0.22; 95% CI: 0.11–0.44), and crack-filling habits (AOR = 0.20; 95% CI: 0.10–0.42). Positive attitude was associated with age 18–40 years (AOR = 2.43; 95% CI: 1.14–5.16), formal education (AOR = 0.67; 95% CI: 0.33–1.36), absence of open defecation (AOR = 0.12; 95% CI: 0.06–0.23), and crack-filling habits (AOR = 0.02; 95% CI: 0.01–0.05).Good practices were associated with age 18–40 years (AOR = 4.30; 95% CI: 1.64–11.10), formal education (AOR = 2.34; 95% CI: 1.31–6.70.), family history of CL (AOR = 0.17; 95% CI: 0.09–0.32), and crack-filling (AOR = 0.14; 95% CI: 0.74–0.28).

## Conclusion

The level of knowledge, attitude and practices towards CL in the study area is very low. Good knowledge, positive attitude and good practice were associated with younger age, formal education, absence of open defecation, and crack-filling behavior. Poor knowledge, attitude and practice of the study communities in the study area emphasize the need to initiate health education and awareness campaigns to reduce cutaneous leishmaniasis.

## Author summary

Cutaneous leishmaniasis prevalent disease transmitted by a vector-borne intracellular protozoan. It is a serious public health concern that affects individuals worldwide and results in serious physical and mental problems. Results from several researchers show that leishmaniasis cases are rising globally, primarily as a result of malnutrition, population movement, inadequate housing, weakened immune systems, low socioeconomic status, and environmental changes. In addition, ecological characteristics, drug resistance, exposure of the human population to the parasite and human's behavior were determinant factors for the spread of Leishmaniasis worldwide. Although cutaneous leishmaniasis is not fatal, it can result in issues, deformity, and permanent scarring, which can lead social stigma. In this study we found that residents' age and educational status

were strongly correlated with communities KAP toward CL. Our findings indicates that there is great a gap among individuals regarding CL. Therefore, our investigation calls for continued and strengthened behavioral change communication and social mobilization related activities.

## Introduction

Leishmaniasis refers to a group of disorders caused by various species of the intracellular protozoa in the genus *Leishmania* [1]. The sickness is caused by parasites from the genus *Leishmania* and the phylum Kinetoplast. About 20 different species of the parasite Leishmania cause leishmaniasis. The transmission of the disease to human is by the bite of infected female sand fly of the genus *Phlebotomus* in the Old World and *Lutzomiya* in the New World respectively [2]. The disease had a profound impact on the lives of 12 million individuals globally, primarily affecting the world's most impoverished population [3]. Globally, about 700,000–1 million new cases of leishmaniasis have been reported annually [4]. There are three main clinical forms of the disease: cutaneous leishmaniasis, visceral leishmaniasis and mucocutaneous leishmaniasis [5].

Cutaneous leishmaniasis is the most common type of leishmaniasis. It is serious public health problem throughout the globe [6]. Globally, approximately 1.5 million new cases of CL are emerging annually and around 350 million people are at risk [7]. The disease causes skin lesions on the exposed parts of the body and mainly caused by *L. major, L. tropica, and L. aethiopica* in Europe, Asia and Africa and *L. mexicana* in the Americas [8]. It primarily affecting the poorest community in low- and middle-income countries [9].The disease has a significant impact on the quality of life and socio-economic status of affected individuals and communities [10].

In Ethiopia, CL is a growing public health problem [11]. It is endemic in the highland areas with an elevation of 1,400–3,175 meters above sea level [12]. Annually, an estimated 20,000–50,000 new cases of CL have been reported in Ethiopia including, 300 CL cases reported in Silti district [6]. Different Leishmania species prevail in different geographical regions and differ by vector and reservoir, *Phlebotomus longipes* and *Phlebotomus pedifer* were identified as the most dominant sand fly species which accounted for CL. in different parts of Ethiopia [13].

*Leishmania aetiopica* is the main cause of CL in Ethiopia causes the most severe of CL [14]. The high incidences of CL have been linked to environmental and behavioral changes and individual risk factors:- Urbanization, deforestation, Movement of endemic area, Decline in social and economic status and Inadequate vector or reservoir control or other human activity associated with sand fly habitat [15]. The disease has a substantial influence on the quality of life and economical status of affected individuals and communities [12].

Rural communities bear the highest disease burden. The lack of health facilities and inaccessibility of appropriate treatments worsen the situation [16]. Mostly vulnerable and poor individuals living in rural communities or those lacking urban services are exposed to the disease. Due to insufficient awareness, they often lack a proper understanding of how to deal with the disease [17]. In endemic areas, prevention and control programs of the CL depend on the existence of a reservoir host, bio-ecological conditions, and diversity of parasite species, making it complicated [18]. Knowing the impact of leishmaniasis and human behaviour surrounding CL is very important to improve its control and treatment [19]. On the other hand, participation and collaboration of the affected population in control plans are essential and represent one of the most effective methods for tackling the prevalence of CL [20].

Therefore, CL control is very important. Because there is still no candidate vaccine with a sufficient and long-lasting effectiveness for public health, CL control depends mainly on early and accurate diagnosis and effective treatment [21]. A timely and definitive diagnosis of CL has great importance for appropriate therapeutic strategy [22].

Ethiopia's National Master Plan emphasizes the importance of community empowerment, integrated disease management, and behavior change communication as key strategies for controlling and ultimately eliminating neglected tropical

diseases (NTDs), including CL. However, despite these strategic priorities, community awareness and engagement in pre-vention and treatment efforts remain limited and inconsistent. The knowledge, attitudes, and practices (KAP) of affected populations vary significantly across geographic and socio-cultural contexts, influenced by factors such as education level, access to health information, traditional beliefs, and the availability of health services. Identifying gaps in KAP is essential for guiding the development of locally adapted community interventions.

Despite the growing burden of CL in Ethiopia, there is a notable lack of community-based KAP studies, particularly in the Kambata zone, which is the focus area of this research. Therefore, this study aimed to assess the communi-ty's knowledge, attitudes, and practices regarding CL. The findings will provide valuable insights into how best to tailor community-based interventions, enhance awareness, and thereby contributing to the national goal of reducing the burden of CL and enhancing the overall effectiveness of Ethiopia's NTD control strategy in the setting.

## Materials and methods

### Study area, design, period, and population

The research was conducted in Kambata zone, situated within the central Ethiopia Regional State, approximately 250 km Southwest of Addis Ababa. It shares borders with Wolaita zone to the South, Dawuro zone to the Southwest, Tem-baro Special Woreda to the West, Hadiya zone to the Northwest, Gurage zone to the North, and Alaba zone to the East. According to Kambata zone agricultural office population report of 2022 the zone has a population of 854,947 (470,386 men and 384,561 women) residing in an area of 1,355.89 km², resulting in a population density of 502.13 individuals per km². Urban dwellers make up 14.36% (97,797) of the population [23]. Durame serving as the administrative center and other significant towns including Shinshicho. The zone's elevated altitude contributes to a temperate to cool climate in the northern region and a tropical climate in the south, with the rainy season typically occurring from April to mid-September, peaking between mid-June and September (Fig 1).

### Geographic location of study area

The study was conducted from February 2024 to August 2024, focusing on exploring the vector life cycle and climate influences. It employed a community-based cross-sectional approach within Kambata zone, targeting all individuals resid-ing in the selected kebeles, the smallest administrative units in Ethiopia that function as neighborhoods or communities composed of several households. The study population included individuals who met specific inclusion criteria: they had to be household heads or members aged 18 years or older, have lived in the district for at least six months, and be willing to participate. Exclusion criteria included individuals with mental impairments or communication difficulties. All individuals interviewed during the research period constituted the study units.

### Sample size determination and sampling techniques

The study determined a sample size of 387 households using the single population proportion statistical formula com-monly applied in health studies. This calculation factored in a 95% confidence level, a 5% margin of error, a 1.5 design effect, and a 10% non-response rate. The proportion (P) of 19% was obtained from a previous report on cutaneous leishmaniasis (CL) knowledge in the Wolaita zone [25], serving as the basis for the estimation of the larger sample size. Employing a multistage sampling technique, four out of the eight woredas in the Kambata zone were randomly selected, each contributing one kebele to the study. The total sample size was distributed proportionally to each kebele based on population-based sampling. A structured list of households in each kebele, obtained from kebele leaders, was used as the sampling frame. Systematic random sampling was then applied to select households, with the initial selection made randomly and subsequent households chosen at regular intervals.

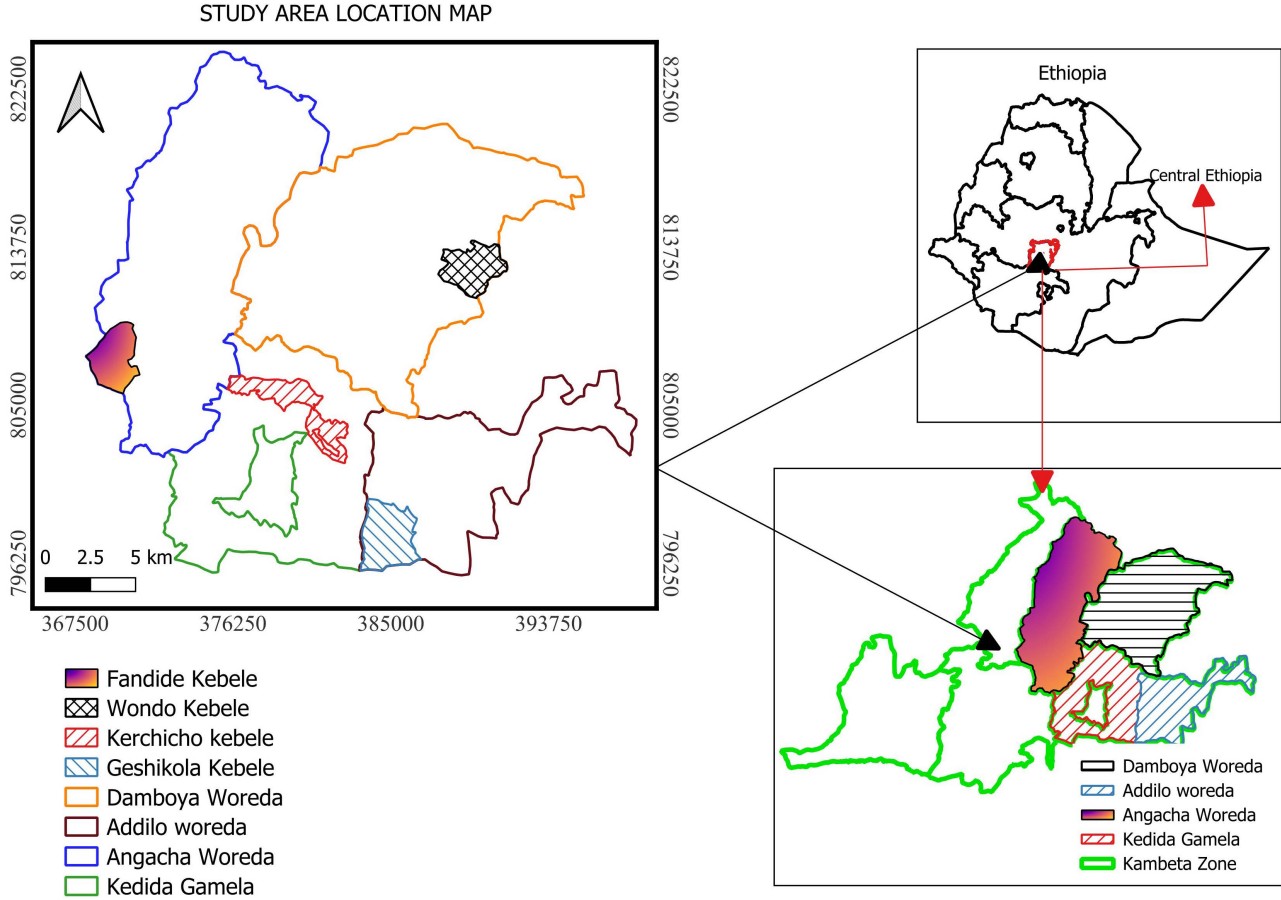

**Fig 1. A location map of the study area (1) [24].**

## Data collection tool, procedure and quality control

The data collection process involved the use of a semi-structured questionnaire, developed based on a thorough review of relevant literature, to gather socio-demographic and other essential information from the study participants. A multidisciplinary team comprising a laboratory technologist, a nurse and a dermatologist were conducted data collection across the selected four villages (a small, rural community where people live close together, often relying on agriculture, livestock, and local resources for their livelihoods), covering a total of 3,940 households. The questionnaire was initially crafted in English and then translated into Amharic and Kambategna. To ensure accuracy and consistency, a language expert, unaware of the translation process, performed a back-translation into English. Field enumerators underwent a comprehensive one-day training session that included an overview of the research objectives, ethical considerations, and data collection procedures. They were also trained to clinically identify cutaneous leishmaniasis and distinguish it from other skin conditions through practical exercises, videos, and visual aids. Prior to full implementation, a pilot test was conducted on a small subset, representing 5% of the target population, to evaluate the clarity and comprehensibility of the questionnaire. Data integrity was closely monitored on a daily basis by the principal investigator and the team leader, who promptly addressed any instances of missing or inconsistent data. All collected data was anonymized and securely stored in a locked cabinet under the principal investigator's supervision to ensure confidentiality and data security.

## Scoring knowledge, attitude, and practice

Participants' knowledge, attitude, and practice were evaluated using a structured scoring system adapted from previous studies [26].

- Knowledge was measured using six items, including awareness of CL, its zoonotic nature, the sand fly vector, transmission, symptoms, and consequences if untreated. Scores ranged from 0 to 6, with scores <3 classified as poor knowledge and ≥3 as good knowledge.

- Attitude was assessed with six items covering perceptions of CL as a health problem, treatability, preventability, stigma, sand fly breeding sites, and movement range. Scores ranged from 0 to 6, with <3 indicating a poor attitude and ≥3 a good attitude.

- Practice was evaluated using five items on prevention methods, treatment preferences, participation in community-based programs, bed net use, and indoor spraying. Scores ranged from 0 to 5, with <3 considered poor practice and ≥3 as good practice

## Data processing and analysis

The data was entered into Epi Data 4.6 and analyzed using SPSS 25.0 software. Descriptive statistics, such as frequencies, percentages, and means, were employed to summarize socio-demographic variables, including categorical variables represented as frequencies and percentages. Sequential bivariate and multivariate logistic regression analyses were performed to investigate the relationships between KAP levels and socio-demographic factors. Variables with a p-value below 0.25 in the bivariate analysis were included in the multivariate model to construct the final model. The forward likelihood ratio method was utilized, and variables with a p-value less than 0.05 were deemed statistically significant, with adjusted Odds ratios (AORs) and 95% confidence intervals (CIs) used to indicate the strength of associations. The results were then presented through figures, tables, and textual representations.

## Results

### Demographic and socio-economic profile of the participants

The data reveal key demographic and socio-economic characteristics of the 387 individuals surveyed, reflecting a 100% response rate. Among the participants, 282 (72.9%) were male and 105 (27.1%) were female. Age distribution indicated that 124 (32%) were aged 18–40, while 263 (68%) were over 40 years old. The majority resided in rural areas (256, 66.1%) compared to urban (131, 33.9%). Occupationally, farmers constituted the largest group at 218 (58.3%), followed by housewives (61, 15.8%) and government employees (53, 13.7%). A significant portion, 278 (71.8%), had formal education, with 253 (65.4%) identifying as Protestant. Notably, 346 (89.4%) owned domestic animals, while 303 (78.3%) had no family history of CL. The data also indicated that 249 (64.3%) of households had latrines, and open defecation was reported by 118 (30.5%). Regarding housing quality, wall material was predominantly mud (213, 55.0%), as was floor material (224, 57.9%), and roofing was mainly wood (178, 46.0%) (Table 1).

### Attitude and knowledge of participants on cutaneous leishmaniasis

The data presented in the Table 2 reveals that 163 participants (42.1%) indicated they had heard of cutaneous leishmaniasis (CL), while 224 participants (57.9%) had not. Only 50 participants (12.9%) knew what zoonosis is, with 113 participants (29.19%) responding negatively and 224 participants (57.9%) indicating they did not know. Additionally, 68 participants (17.57%) reported having heard about sand fly, while 279 participants (72.09%) had not, and 36 participants (9.3%) believed transmission occurred through mosquito bites.

**Table 1. Demographic and socio-economic profile of the participants (N = 387).**

| Variables | Category | Frequency (n) | Percent (%) |
|---|---|---|---|
| Gender | Male | 282 | 72.9 |
| | Female | 105 | 27.1 |
| Age of participants (years) | <18 | 124 | 32 |
| | ≥40 | 263 | 68 |
| Address | Rural | 256 | 66.1 |
| | Urban | 131 | 33.9 |
| Occupation | House wife | 61 | 15.8 |
| | Farmer | 218 | 58.3 |
| | Government employer | 53 | 13.7 |
| | Others | 55 | 14.2 |
| Educational status | Have formal education | 278 | 71.8 |
| | Have no formal education | 109 | 28..2 |
| Religion | Protestant | 253 | 65.4 |
| | Orthodox | 72 | 18.6 |
| | Muslim | 28 | 7.2 |
| | Catholic | 34 | 8.7 |
| | Others | 0 | 0.0 |
| Presence of domestic animals | Yes | 346 | 89.4 |
| | No | 41 | 10.6 |
| Family history of CL | Yes | 84 | 21.7 |
| | No | 303 | 78.3 |
| Population movement | Yes | 68 | 17.6 |
| | No | 319 | 82.4 |
| Presence of latrine | Yes | 249 | 64.3 |
| | No | 138 | 35.7 |
| Crack filling habit | Yes | 82 | 21.2 |
| | No | 305 | 78.8 |
| Exercise open defecation habit | Yes | 118 | 30.5 |
| | No | 269 | 69.5 |
| House wall Material | Mud | 213 | 55.0 |
| | Stone | 119 | 30.7 |
| | Other | 55 | 14,2 |
| House floor material | Mud | 224 | 57.9 |
| | Stone | 130 | 33.6 |
| | Other | 33 | 8.5 |
| Roofing material | Wood | 178 | 46.0 |
| | Stone | 134 | 36.6 |
| | Other | 75 | 19.4 |

Regarding the transmission of CL, 40 participants (10.33%) mentioned sharing sharp materials, 34 participants (8.7%) cited direct contact, 47 participants (12.1%) identified sand flies, and 11 participants (2.8%) mentioned transmission by air droplets, whereas 319 participants (82.42%) admitted they did not know. In terms of recognizing signs of CL, 17 participants (4.3%) identified painful lesions on different sites, 20 participants (5.16%) mentioned painless lesions,59 participants (15.24%) referred to disfiguring skin, and 12 participants (3.1%) mentioned other signs, with 279 participants

**Table 2. Attitude and knowledge of participants on cutaneous leishmaniasis in central Ethiopia.**

| Variables | Category | Frequency | Percent (%) |
|---|---|---|---|
| Have you ever heard CL | Yes | 163 | 42.1 |
| | No | 224 | 57.9 |
| Do you Know what zoonosis is? | Yes | 50 | 12.9 |
| | No | 113 | 29.19 |
| | Don't know | 224 | 57.9 |
| Heard about sand fly | Yes | 68 | 17.57 |
| | No | 279 | 72.09 |
| | By mosquito bite | 36 | 9.3 |
| Transmission of CL | By sharing sharp materials | 40 | 10.33 |
| | Direct contact | 34 | 8.7 |
| | Sand fly | 47 | 12.1 |
| | By air droplets | 11 | 2.8 |
| | Don't know | 319 | 82.42 |
| Do you know Signs of CL | Painful lesion on different sites | 17 | 4.3 |
| | Painless lesion on different sites | 20 | 5.16 |
| | Disfiguring skin | 59 | 15.24 |
| | Others* | 12 | 3.1 |
| | Don't know | 279 | 72.09 |
| Outcome of CL if Un treated | No healing of lesion | 11 | 2.8 |
| | Healing of lesion | 22 | 5.6 |
| | Skin disfiguring | 32 | 8.9 |
| | Don't know | 322 | 83.2 |
| Knowledge (overall) | Good | 70 | 18.1 |
| | Poor | 317 | 81.9 |

**\*Others** (itching, fever, seating, headache, poor appetite, diarrhea, weakness, sleeplessness, dizziness, red eye, weight loss, body swelling).

(72.09%) indicating they did not know. When asked about the outcome of untreated CL, 11 participants (2.8%) stated no healing of lesions, 22 participants (5.6%) mentioned healing of lesions, 32 participants (8.9%) referred to skin disfigurement, and 322 participants (83.2%) admitted they did not know. Overall knowledge about CL was reported as good by 70 participants (18.1%), while 317 participants (81.9%) had poor knowledge of the disease (Table 2).

### Attitudes and beliefs of participants toward cutaneous leishmaniasis

The assessment of community attitudes toward cutaneous leishmaniasis (CL) revealed generally low levels of awareness and uncertainty across key concepts related to the disease. When asked whether CL is a health problem, only 103 participants (26.6%) agreed, while 60 (15.5%) did not consider it a health problem, and the majority, 224 (57.9%), reported that they did not know. Regarding treatability, 73 participants (18.9%) believed CL is treatable, whereas 90 (23.3%) believed it is not; again, more than half of the respondents (224; 57.9%) were unsure. Perceptions about transmission management showed similar trends. Only 58 participants (15.0%) felt that CL transmission is manageable, while 105 (27.1%) believed it is not manageable, and 224 (57.9%) reported that they did not know. Understanding of the social impact of the disease was also limited: 84 participants (21.7%) recognized CL as a stigmatizing disease, while 79 (20.4%) did not, and 224 (57.9%) were uncertain. Knowledge about the sand fly vector was low. Only 65 participants (16.8%) believed that sand

flies breed inside homes, whereas 98 (25.3%) disagreed and 224 (57.9%) did not know. Similarly, 66 participants (17.1%) believed sand flies move long distances, 97 (25.1%) did not believe so, and again 224 (57.9%) did not know. Overall, the attitude assessment showed that only 51 participants (13.2%) demonstrated a good attitude toward CL, while a large majority, 336 participants (86.8%), exhibited poor attitudes (Table 3).

## The practices of study participants in CL prevention and treatment

The survey results show low awareness and poor preventive practices toward cutaneous leishmaniasis (CL). Only 111 participants (28.7%) reported using bed nets to prevent CL, while 31 (8.0%) mentioned avoiding breeding sites and 21 (5.4%) reported spraying chemicals. More than half, 224 participants (57.9%), said they did not know how to prevent CL. Regarding treatment preferences, 63 participants (16.3%) preferred traditional healers, 100 (25.8%) preferred modern medications, and 224 (57.9%) had no preference. Awareness of community-based CL control programs was very limited, with 163 participants (100%) reporting that no program exists, and 224 (57.9%) responding that they did not know of any such program. In terms of preventive practices, 104 participants (26.9%) used bed nets and 59 (15.2%) did not, while 224 (57.9%) did not know. Similarly, only 33 participants (8.5%) reported using indoor insecticides and 130 (30.5%) did not, with another 224 (57.9%) unsure. Overall, just 45 participants (11.6%) demonstrated good preventive practice, whereas 342 (84.4%) showed poor practice (Table 4).

## Factors associated with knowledge domains toward cutaneous leishmaniasis (CL)

Multivariable logistic regression identified several determinants of knowledge about cutaneous leishmaniasis (CL). Sex was significantly associated with knowledge, with males showing higher odds of having good knowledge compared with females (AOR = 1.25, 95% CI: 1.12–2.56, p = 0.01). Age also demonstrated a strong relationship, as participants aged

**Table 3. Attitudes and beliefs of participants toward cutaneous leishmaniasis in central Ethiopia (n = 387).**

| Variables | Categories | Frequency | Percent |
|---|---|---|---|
| Cutaneous leishmaniasis (CL) poses a considerable health challenge. | Yes | 103 | 26.6 |
| | No | 60 | 15.5 |
| | I don't know | 224 | 57.9 |
| Treatment options are available for individuals affected by CL. | Yes | 73 | 18.9 |
| | No | 90 | 23.3 |
| | Don't know | 224 | 57.9 |
| Efforts to manage the transmission of CL have shown promise. | Yes | 58 | 15.0 |
| | No | 105 | 27.1 |
| | Don't know | 224 | 57.9 |
| CL is accompanied by a social stigma that impacts affected individuals. | Yes | 84 | 21.7 |
| | No | 79 | 20.4 |
| | Don't know | 224 | 57.9 |
| Sand flies have been observed to breed within residential settings. | Yes | 65 | 16.8 |
| | No | 98 | 25.3 |
| | Don't know | 224 | 57.9 |
| Sand fly move long distance | Yes | 66 | 17.1 |
| | No | 97 | 25.1 |
| | Don't know | 224 | 57.9 |
| Attitude of participant regarding CL | Good | 51 | 13.2 |
| | Poor | 336 | 86.8 |

**PLOS Neglected Tropical Diseases**

**Table 4. Practices of study participants in CL prevention and treatment in Central Ethiopia (n = 387).**

| Variables | Categories | Frequency | Percentage (%) |
|---|---|---|---|
| How can you prevent CL | Using bed net | 111 | 28.7 |
| | Avoiding breeding site | 31 | 8.0 |
| | Spraying chemicals | 21 | 5.4 |
| | Don't know | 224 | 57.9 |
| Treatment preferences | Traditional healers | 63 | 16.3 |
| | Modern medications | 100 | 25.8 |
| | None | 224 | 57.9 |
| Community-based CL control program | Yes | 0 | 0.0 |
| | No | 163 | 100.0 |
| | Don't know | 224 | 57.9 |
| Use bed net | Yes | 104 | 26.9 |
| | No | 59 | 15.2 |
| | Don't know | 224 | 57.9 |
| Use insecticides indoor | Yes | 33 | 8.5 |
| | No | 130 | 30.5 |
| | Don't know | 224 | 57.9 |
| Over all practice | Good | 45 | 11.6 |
| | Poor | 342 | 84.4 |

18–40 years were more likely to have good knowledge than those older than 40 years (AOR = 4.47, 95% CI: 2.07–9.67, p = 0.01). Formal education was an important predictor, with individuals who had formal schooling being more likely to exhibit good knowledge compared with those without education (AOR = 3.67, 95% CI: 2.18–8.75, p = 0.06). Additionally, family history of CL significantly influenced knowledge levels; participants without a family history had lower odds of good knowledge compared with those who had such a history (AOR = 0.22, 95% CI: 0.11–0.84, p = 0.01). The habit of filling cracks in house walls was also a significant factor, as individuals who did not practice filling cracks were less likely to have good knowledge (AOR = 0.20, 95% CI: 0.10–0.42, p = 0.01) (Table 5).

### Factors associated with attitudes domains toward Cutaneous Leishmaniasis

A multivariable logistic regression analysis identified several significant factors influencing participants' attitudes towards Cutaneous Leishmaniasis. Age was a key predictor, with individuals aged 18–40 years demonstrating a greater propensity for positive attitudes towards CL compared to those over 40 years of age AOR = 2.43, 95% CI: 1.14-5.16, p = 0.02). Additionally, formal education was strongly associated with favorable attitudes; participants with formal education had higher odds of possessing positive attitudes towards CL than those without (AOR = 2.67, 95% CI: 3.33-8.36, p = 0.01). The practice of filling cracks in household walls was also significantly linked to attitudes, as individuals who engaged in this practice were more likely to exhibit positive attitudes towards CL prevention and control (AOR = 3.02, 95% CI: 1.01-9.05, p = 0.01). Furthermore, open defecation behavior emerged as a significant factor; individuals who abstained from open defecation were more likely to hold favorable attitudes towards CL compared to those who did practice it (AOR = 1.02, 95% CI: 1.06-5.23, p = 0.01) (Table 6).

### Preventive practice factors associated with cutaneous leishmaniasis

The multivariable logistic regression analysis revealed several factors significantly associated with preventive practices toward CL. Age was a notable predictor, with participants aged 18–40 years being more likely to engage in effective

**Table 5. Logistic regression analysis of factors associated with knowledge about cutaneous leishmaniasis.**

| Variables | Category | Knowledge About CL | | COR (95%CI) | P value | AOR (95%CI) | P value |
|---|---|---|---|---|---|---|---|
| | | Good n (%) | Poor n (%) | | | | |
| Sex | Male | 63(90) | 219(69.1) | 1.15(1.02,3.83) | 0.03 | 1.253(1.117, 2.56) | 0.01 |
| | Female | 10(10.0) | 98(30.9) | 1 | | 1 | |
| Age | 18-40 | 8(11.4) | 116(36.6) | 2.53 (1.94, 6.77) | 0.07 | 4.473 (2.07,9.67) | 0.01 |
| | > 40yrs | 62(88.6) | 201(63.4) | 1 | | 1 | |
| Address | Rural | 46(65.7) | 210(66.2) | 1 | | 1 | |
| | Urban | 24(34.3) | 107(33.8) | 4.80(1.124,18.55) | 0.023 | 1.02 (0.59,1.77) | 0.93 |
| Formal education | Yes | 60 (85.7) | 219(68.8) | 2.08(1.23,5.40) | 0.01 | 3.67 (2.18,8.75) | 0.06 |
| | No | 10(14.3) | 99(31.2) | 1 | | 1 | |
| Family history | Yes | 31(44.3) | 53(16.7) | 1 | | 1 | |
| | No | 39(55.7) | 264(83.3) | 0.22(0.098,0.477) | 0.01 | 0.22 (0.11,0.84) | 0.01 |
| Movement history | Yes | 36(51.4) | 206(65.0) | 2.23(1.80,6.16) | 0.12 | 1.18(0.58,2.38) | 0.65 |
| | No | 34(48.6) | 111(35.0) | 1 | | 1 | |
| Habit of filling cracks | Yes | 34(48.6) | 48(15.1) | 1 | | 1 | |
| | No | 36(51.4) | 269(84.9) | 0.191(0.08,0.46) | 0.01 | 0.20(0.0.10,0.42) | 0.01 |

**Table 6. Logistic regression analysis of factors associated with attitudes toward cutaneous leishmaniasis.**

| Variables | Category | Attitude toward CL | | COR (95%CI) | P value | AOR (95%CI) | P value |
|---|---|---|---|---|---|---|---|
| | | Good n (%) | Poor n (%) | | | | |
| Age | 18-40 | 9 (17.6) | 115 (34.2) | 3.41 (1.53,21.81) | 0.12 | 2.43 (1.14,5.16) | 0.02 |
| | > 40yrs | 42 (82.4) | 221 (65.8) | | | 1 | |
| Sex | Male | 38 (74.5) | 244 (72.6) | 0.15 (0.01, 0.89) | 0.14 | 0.91 (0.46,1.78) | 0.78 |
| | Female | 13 (25.5) | 92 (27.4) | 1 | | 1 | |
| Formal education | Yes | 40 (78.4) | 238 (70.8) | 0.12 (0.07,0.87) | 0.13 | 2.67 (3.33,8.36) | 0.01 |
| | No | 11 (21.6) | 98 (29.1) | 1 | | 1 | |
| Movement history | Yes | 8 (15.7) | 60 (17.9) | 3.017 (1.66,13.79) | 0.15 | 1.17 (0.52,2.61) | 0.71 |
| | No | 43 (84.3) | 276 (82.1) | 1 | | 1 | |
| Habit of crack filling | Yes | 44 (86.3) | 38 (11.3) | 2.01 (1.02,7.06) | P<0.01 | 3.02 (1.01,9.05) | 0.01 |
| | No | 7 (13.7) | 298 (88.7) | 1 | | 1 | |
| Open defecation habit | Yes | 37 (72.5) | 81 (24.1) | 1 | | 1 | |
| | No | 14 (27.5) | 255 (75.9) | 0.169 (0.05,0.58) | 0.01 | 1.02 (1.06,5.23) | 0.01 |
| Roofing material | Mud | 30 (58.8) | 148 (44.0) | 1 | | | |
| | Stone | 16 (31.4) | 118 (35.1) | | | | |
| | Metal | 5 (9.8) | 70 (20.8) | 0.16 (0.02,1.08) | 0.06 | 0.53 (0.19,1.50) | 0.23 |

preventive measures compared to those over 40 years (Adjusted Odds Ratio [AOR] = 4.27, 95% Confidence Interval [CI]: 1.64-11.10, p = 0.01). Additionally, formal education emerged as a significant factor; individuals with formal education displayed higher odds of adopting appropriate preventive practices than those without formal education (AOR = 2.34, 95% CI: 1.31-6.69, p = 0.02) (Table 7).

## Discussion

The survey findings provide important insights into the demographic and socio-economic characteristics of participants, as well as their knowledge, attitudes, and practices related to cutaneous leishmaniasis (CL) in Central Ethiopia. With a

**Table 7. Logistic regression analysis of factors associated with preventive practices regarding cutaneous leishmaniasis.**

| Variables | Category | Practice toward CL | | COR (95%CI) | P value | AOR (95%CI) | P value |
|---|---|---|---|---|---|---|---|
| | | Good n (%) | Poor n (%) | | | | |
| Age | 18-40 | 5 (11.1) | 119 (34.8) | 2.67 (1.68,10.47) | 0.16 | 4.27 (1.64,11.10) | 0.01 |
| | > 40yrs | 40 (88.9) | 223 (65.2) | 1 | | 1 | |
| Formal education | Yes | 37 (82.2) | 241 (70.5) | 1.05 (1.02,2.93) | 0.10 | 2.34 (1.31,6.69) | 0.02 |
| | No | 8 (17.8) | 101 (29.5) | 1 | | 1 | |
| Family history of CL | Yes | 25 (55.6) | 59 (17.3) | 0.16 (0.06,0.40) | 0.01 | 3.17 (1.09,8.32) | 0.01 |
| | No | 20 (44.4) | 283 (82.7) | 1 | | 1 | |
| Crack filling habit | Yes | 26 (57.8) | 56 (16.4) | 0.09 (0.03,0.29) | 0.01 | 0.84 (0.14, 0.98) | 0.01 |
| | No | 19 (42.2) | 286 (83.6) | 1 | | 1 | |

100% response rate, the data represents diverse community backgrounds that may influence awareness, perception, and preventive behavior toward the disease. Furthermore, the study indicated that cutaneous leishmaniasis remains one of the most neglected tropical diseases in Ethiopia, and its findings highlight critical gaps that continue to hinder effective prevention and control efforts [27]. Peoples' awareness and perception towards CL have its own impact to prevent the disease in endemic areas [28]. This study assessed knowledge, attitude and practices concerning CL among rural and urban communities living in Kambata zone. Less than half (42.1%) participants had heard about the diseases, which is consistent with study conducted in Saud Arabia Tubaik [29], in which significant portion of the study population were unaware of CL. But our result were much lower than studies conducted in Tigrai Northern Ethiopia [6], in Kutaber district, North east Ethiopia [30], in Delanta district, North east Ethiopia [28], in Volta Region of Ghana [19] and Iran [20] were 99%, 99%, 76.8%, 95.5%, and 90.4% of the study participants have knowledge on CL respectively.

Mammalian reservoir hosts are responsible for maintaining in a population of an infectious agent in Ethiopia for a long period of time [31]. Almost, all participants have different types of domestic animals, but 87.1% of them didn't recognize the zoonotic nature of the disease. This is in line with findings reported from Wolaita zone where 87.4% of the respondents didn't recognized that CL is a zoonotic disease [25]. Very few, 17.57% of participants have heard about sand fly, among them a few 12.1% study participants correctly responded that the sand fly as transmit agent for the disease. This finding is in agreement with a study conducted in Volta region of Ghana [19] and southwestern Yemen [32] were 19.8% and 14.8% of the participants knew that a fly transmit disease respectively. Our findings were lower than the findings conducted in North West Ethiopia [33] in Pakistan [15], in Southwest Iran [34] and in Saudi Arabia [29] were 58.09%, 53.5%, 65.6% and 47.5% of participants knew that sand fly transmit diseases respectively. Very few, 15.24% of participants responded that skin disfigurement was a marker of CL. These findings were lower than the findings conducted, in Saudi Arabia [29] in Iran [20], in South western Yemen [32] and in North West Ethiopia [33] were 43.6%, 84%, 82.0% and 72.3% of respondents correctly identified the symptoms of CL respectively.

Overall, 18.1% of participants in our study area had good knowledge about CL. These result is in line with study conducted in Wolaita zone in which 19% of participants had good knowledge [25]. Our findings were lower than study conducted in Kutaber district [30], in Delanta district [28] and in Southwestern Yemen [32] were 47.5%,27.6% and 22.3% the of the participants had good knowledge about CL respectively. This knowledge gap among study participants might be due to the high endemic nature of the disease in different locations and due to lack of community-based health education in the study area. In general, our results show that participants' knowledge in the subject area is greatly lacking, these highlighting the need to improve public health education.

Among study participants 26.6% of participants thought that CL is health problem in the study area. These were lower than finding conducted in Kutaber district [30] in Wolaita zone [25], in Tigrai [6], in north west Ethiopia [33] and in Delanta district [28] were 45.4%, 71.6%, 78%,77.9% and 61.1% of participants thought that CL was a health problem respectively.

Regarding the treatability of CL, only 18.9% of the study participants believe that CL is a treatable disease. This finding was lower than findings conducted in Siri lanka [35], in Saudi Arabia [29], in southwestern Yemen [32], in Wolaita zone [25] and in Delanta district [28] were over 75%, 59%, 82.8%, 37.2% and 55.5% of the participants responded that the disease is treatable respectively.

More than half (85.0%) of study participants believe that the diseases transmission was not preventable. It was consistent with findings in Tubaik Saudi Arabia [29] and in southwestern Yemen [32] showed that the majority of study participants had poor attitude of CL management. Our finding was lower than study conducted in Delanta district [28], in which majority of participants believed that CL is a preventable disease.

In this study 21.7% of participants believe CL as stigmatizing disease. Different study conducted in different locations indicated that people have negative perception toward CL. In studies conducted in Wolaita zone, 43.8% of the respondents [25], in southwest Yemen half of the participants had a negative attitude towards CL [32]. Regarding the breeding site of the fly, 25.5% of participants believe that the sand fly does not breed inside the home. It is consistent with study conducted in Pakistan in which both the urban and rural population were unable to give accurate description of breeding breeding sites [15]. Our findings was lower than findings conducted in Tabuk, Saudi Arabia in which 38.5% of rural residents and 19.3% of urban residents correctly identified the breeding places of sand flies [29].

Generally, from all study participants only 13.2% participants had a good attitude about CL. This finding was lower than the finding conducted in Kutaber district [30], and in Delanta district [28] in which 54.1% and 34.5% of the study participants had a positive attitude toward CL respectively. Therefore, the overall poor level attitude of CL revealed by the current study could be a direct consequence of this disease being neglected by health policy makers and public health professionals.

Regards to prevention mechanisms of CL 28.7% participants stated that sleeping under bed net is the best way to prevent the disease. It was comparable with findings conducted in Pakistan [15] and North West Ethiopia [36] in which 25.8% and 31.6% participants indicated that bet nets as preventive means for CL respectively. And lower than findings conducted in southwestern Yemen [32] and Iran [34] in which 40.1% and 50% participants indicated that bet nets as preventive means for CL respectively.

Among the study participants 32.6% of the participants responded that modern medications are the best option in order to treat CL. This aligns with findings from North west Ethiopia [33], Kutaber district [30], Saudi Arabia [29] and south west Iran [34] were 27.94%, 40%, 54% and 41.5% of participants indicated modern medicines as their best treatment option respectively. While our finding was higher than findings conducted in Wolaita zone [25], Amhara region [33] and Tigray region [6] were 77%, 68.3% and 90% patients remain dependent on traditional therapies respectively. In localities like the present study area, where modern CL treatment is unavailable, CL patients remain dependent on traditional therapies.

None of the participants participated on community-based CL control program. This exhibit similar findings with study conducted in Volta Village of Ghana in which 97.3% of the participant reported that they had never had any formal education on leishmaniasis in the community [19]. Therefore, these findings emphasize the need to implement a health education and awareness campaign to reduce the risk of CL infection. In this study, the knowledge of the participants about CL was significantly associated with Age, sex, educational status, family history and habit of filling cracks. The male participants had better knowledge as compared to their female counterparts. In our country, male individuals were involved in many outdoor activities than females. Therefore, the probability of having good knowledge about the disease is more among males than females. The outcome of the multivariate logistic regression shows that, age education level, open defecation habit and crack filling habit were significant determinants of good attitude about CL. Because adult educated individuals have more chances to get more health-related information from different sources than older ones.

Our final analysis on association of different factors with practice toward CL indicates that, age, education status, family history and crack filling habit of individuals were significantly associated with the overall practice of the respondents. There were increased odds of having good practice among participants with age category between 18–40 years with because

this age class includes active individuals who have good practice to prevent the disease than older. Thus, to prevent the spreading of disease to a non- endemic for CL it is important to increase awareness activities through enhancing health related information.

## Strength and limitation of the study

Limitations of this study include limited number of open-ended questions which would have helped the respondents to add more information to the provided questions, the face-to-face interview method of the data collection might have predisposed the respondents to social desirability bias. Despite these limitations, this study presented valuable findings to emphasize health education campaigns and future CL prevention and control plans in the study area. Hence, the findings of this study should be generalized to the whole population of the study area.

## Conclusion

With the increase in human populations and lack of employment and civic facilities in rural areas, migration to large urban hubs is increasing not only in developing countries. The situation leads to decrease in agricultural lands, uncontrolled urbanization with poor to no facilities of sewerage and hygiene consequently increasing breeding places of insects and onset of arthropod borne health issues. In general our findings showed that the level of knowledge, attitude and practices towards CL prevention were low. Concerning disease control, the people's attitude towards complete cure of the disease, treatability of the disease and control of the disease through community participation were favorable. Our findings indicates that there is great a gap among individuals regarding CL. Therefore, our investigation calls for continued and strengthened behavioral change communication and social mobilization related activities.

## Supporting information

**S1 File. Questionnaire used for the data collection.**
(DOCX)

## Acknowledgments

First of all, the authors would like thanks to all participants who were volunteer to participate in this study. We deeply express our appreciation to Wolaita Sodo University and Jimma University for their valuable support in providing these chance. The authors also wish to thank the Kambata zone Health bureau, for their invaluable support through the whole process and we are grateful to the supervisors and data collectors who have committed themselves throughout the study period.

## Author contributions

**Conceptualization:** Tagesech Yohannes, Getenet Beyene, Teketel Ermias Geltore, Zeleke Mekonnen, Daniel Geleta.

**Data curation:** Tagesech Yohannes, Getenet Beyene, Teketel Ermias Geltore, Zeleke Mekonnen, Daniel Geleta.

**Formal analysis:** Tagesech Yohannes, Getenet Beyene, Teketel Ermias Geltore, Zeleke Mekonnen.

**Funding acquisition:** Tagesech Yohannes, Getenet Beyene, Teketel Ermias Geltore, Zeleke Mekonnen, Daniel Geleta.

**Investigation:** Tagesech Yohannes, Getenet Beyene, Teketel Ermias Geltore, Zeleke Mekonnen.

**Methodology:** Tagesech Yohannes, Getenet Beyene, Teketel Ermias Geltore, Zeleke Mekonnen, Daniel Geleta.

**Project administration:** Tagesech Yohannes, Getenet Beyene, Teketel Ermias Geltore, Zeleke Mekonnen, Daniel Geleta.

**Resources:** Tagesech Yohannes, Getenet Beyene, Teketel Ermias Geltore, Zeleke Mekonnen, Daniel Geleta.

**Software:** Tagesech Yohannes, Getenet Beyene, Teketel Ermias Geltore, Zeleke Mekonnen, Daniel Geleta.

**Supervision:** Tagesech Yohannes, Getenet Beyene, Teketel Ermias Geltore, Zeleke Mekonnen.

**Validation:** Tagesech Yohannes, Getenet Beyene, Teketel Ermias Geltore, Zeleke Mekonnen, Daniel Geleta.

**Visualization:** Tagesech Yohannes, Getenet Beyene, Teketel Ermias Geltore, Zeleke Mekonnen, Daniel Geleta.

**Writing – original draft:** Tagesech Yohannes.

**Writing – review & editing:** Tagesech Yohannes, Getenet Beyene, Teketel Ermias Geltore, Zeleke Mekonnen, Daniel Geleta.

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
