## [Decision Letter · Decision Letter 0]

10 Nov 2025

Response to Reviewers
Revised Manuscript with Track Changes
Manuscript

Peter Kima

Guest Editor

Shaden Kamhawi

co-Editor-in-Chief

Paul Brindley

co-Editor-in-Chief

**Additional Editor Comments:**
**Journal Requirements:**

At this stage, the following Authors/Authors require contributions: Tagesech Yohannes. Please ensure that the full contributions of each author are acknowledged in the "Add/Edit/Remove Authors" section of our submission form.

3) Some material included in your submission may be copyrighted. According to PLOSu2019s copyright policy, authors who use figures or other material (e.g., graphics, clipart, maps) from another author or copyright holder must demonstrate or obtain permission to publish this material under the Creative Commons Attribution 4.0 International (CC BY 4.0) License used by PLOS journals. Please closely review the details of PLOSu2019s copyright requirements here: PLOS Licenses and Copyright. If you need to request permissions from a copyright holder, you may use PLOS's Copyright Content Permission form.

Potential Copyright Issues:

i) Figure 1. Please (a) provide a direct link to the base layer of the map (i.e., the country or region border shape) and ensure this is also included in the figure legend; and (b) provide a link to the terms of use / license information for the base layer image or shapefile. We cannot publish proprietary or copyrighted maps (e.g. Google Maps, Mapquest) and the terms of use for your map base layer must be compatible with our CC BY 4.0 license.

4) We note that your Data Availability Statement is currently as follows: "All relevant data are within the manuscript and its SupportingInformation files.". Please confirm at this time whether or not your submission contains all raw data required to replicate the results of your study. Authors must share the “minimal data set” for their submission. PLOS defines the minimal data set to consist of the data required to replicate all study findings reported in the article, as well as related metadata and methods (https://journals.plos.org/plosone/s/data-availability#loc-minimal-data-set-definition).

5) Please provide a detailed Financial Disclosure statement. This is published with the article. It must therefore be completed in full sentences and contain the exact wording you wish to be published.

1) Please clarify all sources of financial support for your study. List the grants, grant numbers, and organizations that funded your study, including funding received from your institution. Please note that suppliers of material support, including research materials, should be recognized in the Acknowledgements section rather than in the Financial Disclosure

2) State the initials, alongside each funding source, of each author to receive each grant. For example: "This work was supported by the National Institutes of Health (####### to AM; ###### to CJ) and the National Science Foundation (###### to AM)."

3) State what role the funders took in the study. If the funders had no role in your study, please state: "The funders had no role in study design, data collection and analysis, decision to publish, or preparation of the manuscript."

4) If any authors received a salary from any of your funders, please state which authors and which funders..

**Reviewers' comments:**

**Key Review Criteria Required for Acceptance?**

**Methods**

-Are the objectives of the study clearly articulated with a clear testable hypothesis stated?

-Is the study design appropriate to address the stated objectives?

-Is the population clearly described and appropriate for the hypothesis being tested?

-Is the sample size sufficient to ensure adequate power to address the hypothesis being tested?

-Were correct statistical analysis used to support conclusions?

-Are there concerns about ethical or regulatory requirements being met?

Reviewer #1: (No Response)

Reviewer #2: The above criteria for acceptance were met except for the sample. However, the sample used for the study does not represent the current population of the study site. A 2007 census was used to obtain the sample studied.

**Results**

-Does the analysis presented match the analysis plan?

-Are the results clearly and completely presented?

-Are the figures (Tables, Images) of sufficient quality for clarity?

Reviewer #1: (No Response)

Reviewer #2: A few data such as house wall material, floor and roofing material in Table 2 were not discussed at all. One would like to see a discussion of the relevance of such data in the epidemiology of CL in the focus

**Conclusions**

-Are the conclusions supported by the data presented?

-Are the limitations of analysis clearly described?

-Do the authors discuss how these data can be helpful to advance our understanding of the topic under study?

-Is public health relevance addressed?

Reviewer #1: (No Response)

Reviewer #2: The conclusions are supported by the data presented. However, as mentioned above the information used to calculate the sample size is questionable.

**Editorial and Data Presentation Modifications?**

Reviewer #1: (No Response)

Reviewer #2: (No Response)

**Summary and General Comments**

Reviewer #1: This paper is part of a crucial study aimed at assessing the understanding and involvement of the population in the Kambata Zone of Ethiopia regarding cutaneous leishmaniasis, contributing to other KAP (Knowledge, Attitude, and Practice) studies already conducted in other regions of Africa. However, I believe that some corrections to the paper should be made for publication. Therefore, I suggest that the authors review the following points:

Introduction

- I suggest the authors revise the introduction, as while informative, the text is somewhat disjointed. The authors describe the disease's epidemiology from a macro perspective (globally), mention Ethiopia, revisit the prevalence in the Old and New Worlds, and then discuss the epidemiology in Ethiopia.

- I also suggest that authors use references from the WHO and other manuals on neglected diseases throughout the text.

- I also suggest the authors include the species and vector that causes CL in this area and the number of cases in the Kambata region, if available.

- They also discuss the specific signs and symptoms, including whether it causes localized lesions or can cause a diffuse form of the disease.

Line 59: I ask authors to correct the disease "Elephantiasis" to "Cutaneous Leishmaniasis or Leishmaniasis," depending on how they would like to conduct the text.

Line 60: I suggest authors add references to the WHO or other neglected disease panels, as these are the organizations that commonly publish data on the number of cases and deaths.

- I suggest authors confirm whether there are 20 species of Leishmania pathogenic to humans or whether there are more than..., as according to the WHO, leishmaniases are a group of diseases caused by protozoan parasites from more than 20 Leishmania species.

Line 62: I suggest the authors correct the spelling of "Lutuzomiya" to "Lutzomyia".

Methods

Line 122: Is there a specific reason the study was conducted during this period? Vector dispersal, rainy season, higher frequency of cases between February and August. If so, I suggest the authors add this information to the text.

Line 161: Data quality control: I suggest the authors include the questionnaire used as supplementary material.

Results:

- When asked about CL, were study participants familiar with or had they ever heard of terms like "Leishmania" or "Leishmaniasis"? Furthermore, did participants mention local/popular names that could be correlated with diseases? If so, I suggest the authors add this information to the results/discussion.

- Figure 1: I suggest the authors add a caption to the figure.

Furthermore, I suggest that the authors include a more informative map, with boundaries and names of the regions on the Kambata Zone map. In addition to informing the reference or program in which the map was created.

- Table 2: Under occupation, what other professions does "Other" fall under?

- Table 3: Participants knew little about the vector, according to the data presented. Did the authors ask participants about popular names for the vector? In rural areas of Brazil, the population knows the sandfly as mosquito palha and/or cangalha, which often facilitates identification of the vector by the population, who are already familiar with the vector but do not recognize the scientific nomenclature.

Tables 4 and 5: The values for "Don't know," "None," and "Neither" in the tables are all the same. Does this mean that all 224 participants were unaware of all the requested information? I suggest the authors confirm these values.

Reviewer #2: The manuscript reports data on KAP toward CL in an endemic area of Ethiopia. These data are relevant for the development and implementation of a control program in the focus. However, the manuscript cannot be published as submitted. I think that the aspects listed below should be taken into consideration in the revision for final acceptance.

- The language should be improved. For example, Line 59, the authors wrote elephantiasis talking about leishmaniasis. In addition, line 60, they have Leishmania that is neither italicized nor in bold. This is true for many genus or species names throughout the text that are neither italicized nor in bold. Line 70, they have leishmaniasis with upper case L.

- In the methods, the authors mentioned using 2007 Census data (Line 116). These data are 17 years old. I am not sure that the sample used in this study represents the current population of the study area. The authors must address this for the MS to be relevant.

- Sampling techniques, the authors should define for non-Ethiopians words such as Angecha and Kebele. In the text, they also use village.

- Results: Line 200, they mention the religion of the participants. However, the discussion does not take into account the religious beliefs of respondents. A few data such as house wall material, floor and roofing material in Table 2 were not discussed at all. One would like to see a discussion of the relevance of such data in the epidemiology of CL in the focus

PLOS authors have the option to publish the peer review history of their article (what does this mean? ). If published, this will include your full peer review and any attached files.

**Do you want your identity to be public for this peer review?** For information about this choice, including consent withdrawal, please see our Privacy Policy .

Reviewer #1: No

Reviewer #2: No

**Figure resubmission:**
---

## [Editor Report · Decision Letter 1]

8 Dec 2025

Dear PhD candidate Yohannes,

We are pleased to inform you that your manuscript 'Knowledge, attitude and practice of community toward Cutaneous Leishmaniasis in Kambata   Zone, Central Ethiopia' has been provisionally accepted for publication in PLOS Neglected Tropical Diseases.

Best regards,

Peter E Kima

Guest Editor

Hira Nakhasi

Section Editor

Shaden Kamhawi

co-Editor-in-Chief

Paul Brindley

co-Editor-in-Chief

---

## [Editor Report · Acceptance letter]

Dear PhD candidate Yohannes,

We are delighted to inform you that your manuscript, "Community knowledge, attitudes, and practices toward cutaneous leishmaniasis in central Ethiopia," has been formally accepted for publication in PLOS Neglected Tropical Diseases.

Best regards,

Shaden Kamhawi

co-Editor-in-Chief

Paul Brindley

co-Editor-in-Chief
